# Combined Treatment of Parts Produced by Additive Manufacturing Methods for Improving the Surface Quality

Sergey Grigoriev , Alexander Metel * , Marina Volosova, Yury Melnik and Enver Mustafaev

Department of High-Efficiency Processing Technologies, Moscow State University of Technology "STANKIN", 1 Vadkovsky lane., Moscow 127055, Russia
* Correspondence: a.metel@stankin.ru; Tel.: +7-903-246-43-22

**Abstract:** To improve the quality of a part manufactured by the additive method, it is necessary to eliminate the porosity and high roughness of its surface, as well as to deposit a coating on it. For this purpose, in the present work, we studied the combined processing in a gas discharge plasma of complex shape parts obtained by the additive manufacturing method, which includes explosive ablation of surface protrusions when voltage pulses are applied to the part immersed in the plasma; polishing with a concentrated beam of fast neutral argon atoms at a large angle of incidence on the surface of the part, and magnetron deposition of a coating on it with assistance by fast argon atoms. Combined processing made it possible to completely get rid of porosity and reduce the surface roughness from Ra ~ 5 μm to Ra ~ 0.05 μm.

**Keywords:** roughness; surface protrusions; explosive ablation; polishing with fast atoms; angle of incidence; coating deposition

## 1. Introduction

Additive manufacturing reduces material costs and production time, and makes it possible to obtain products of complex shape, which are extremely difficult to manufacture by other methods. However, in order to introduce additive manufacturing into the industry, it is first necessary to improve the quality of manufactured parts. Currently, with any 3D printing technology, the surface of additively manufactured parts is porous and extremely rough. In addition, the choice of materials suitable for additive manufacturing is limited, and this dictates the need for coating deposition.

The requirements for the technology of polishing the surface of parts and deposition of coatings depend on the intended application of the parts, their material and printing technology. Mechanical polishing, which, depending on the size of the abrasive used, can significantly reduce surface roughness, is relatively cheap and can provide a fairly uniform surface structure. However, it cannot be used to process the internal cavities of parts with a developed surface relief. In addition, after the mechanical polishing, abrasive particles remain in the surface layer of the material.

Electrochemical etching makes it possible to polish the inner surfaces of complex shape parts and reduces the surface roughness to Ra = 0.01 μm [1]. However, this method requires the disposal of aggressive etch products, which pose a potential hazard to the environment and represent the main obstacle to using this method in the development of new technologies.

Laser-plasma polishing provides deep penetration and volumetric vaporization of the part material, as well as a significant simplification of processing control [2,3]. Its disadvantages are the local effect of the laser beam and the need to create a protective atmosphere that prevents the material from oxidizing during the polishing process. A serious problem is the fact that when processing a part of a complex shape, some of its sections block the access of the laser beam to the surface of other sections.

Polishing the surface of metal products obtained by additive manufacturing with high-current pulsed electron beams makes it possible to almost completely get rid of porosity and reduce the roughness Ra from tens to units of micrometers [4]. However, the electron beam does not reduce the roughness Ra to less than 1 μm, and therefore cannot provide the highest surface finish class.

Systematic studies of the ion beam polishing [5–7] determined that the ion beam energy and incident angle are the main factors affecting the removal of surface protrusions. Research shows that the relationship between the incident angle and the sputtering yield is nonlinear under different energies. New possibilities for polishing dielectric parts include sources of broad, fast, neutral atom beams [8–10].

This paper presents the results of a study of combined processing in a gas discharge plasma of parts obtained by the additive manufacturing method, including:

- explosive ablation of surface protrusions when high-voltage pulses are applied to a part immersed in plasma;
- polishing with a concentrated beam of fast neutral argon atoms at a large angle of incidence on the part surface;
- magnetron deposition on the part of a coating assisted by fast argon atoms.

## 2. Materials and Methods

### 2.1. Experimental Setup

To carry out the experiments, 8-cm-long hollow cylindrical samples with an outer diameter of 5 cm and an inner diameter of 4 cm were fabricated by additive technology. Fine powder for additive manufacturing, obtained by gas atomization from a heat-resistant CoCrMo alloy, was chosen as the starting material for their manufacture. The samples were fabricated at an industrial EOS M 400 additive manufacturing facility using selective laser melting technology [11,12]. Measurements using a HOMMEL TESTER T8000 profilometer by Hommelwerke GmbH (Germany) showed that the surface roughness of the fabricated samples amounted to Ra ~ 5 μm. The measurement conditions were established in accordance with PN-EN ISO 3274:2011 and PN-EN ISO 4288:2011 and are presented in Table 1.

**Table 1.** Surface roughness measuring conditions.

| Type of Profilometer | Hommel Tester T8000 (Hommelwerke GmbH, Germany) |
|---|---|
| Stylus type | TKU 300 |
| Tracing length | $lt = 2.0$ mm |
| Evaluation length | $ln = 1.5$ mm |
| Sampling length | $lr = 0.25$ mm |
| Evaluation width (3D measurements) | $l = 1.0$ mm |
| Number of stylus passes (3D measurements) | 201 |
| Distance between stylus tracks (3D measurements) | 5 μm |
| Stylus tip radius | $r_{tip} = 2$ μm |
| Stylus tip angle | 90° |
| Tracing speed | $v_t = 0.05$ mm/s |
| Long-wave profile filter (cutoff) | $\lambda_c = 0.25$ mm |
| Measuring range | ±80 μm |
| Gaussian digital filter | 80 μm |
| Software | TURBO ROUGHNESS, Hommel Map Expert 4.1. |

Figure 1 shows a diagram of an experimental setup for the combined processing of complex-shape parts. It consists of two water-cooled vacuum chambers. A plasma ion emitter is formed in a cylindrical 50-cm-diameter and 21-cm-long chamber connected with a rectangular 75-cm-long, 14-cm-high, and 40-cm-wide chamber. A rotating sample holder is

located in the rectangular chamber. The holder is isolated from the chamber and connected to the negative pole of a high-voltage pulse generator. The angle between the axis of the holder rotation and the axis of the camera amounts to 15°.

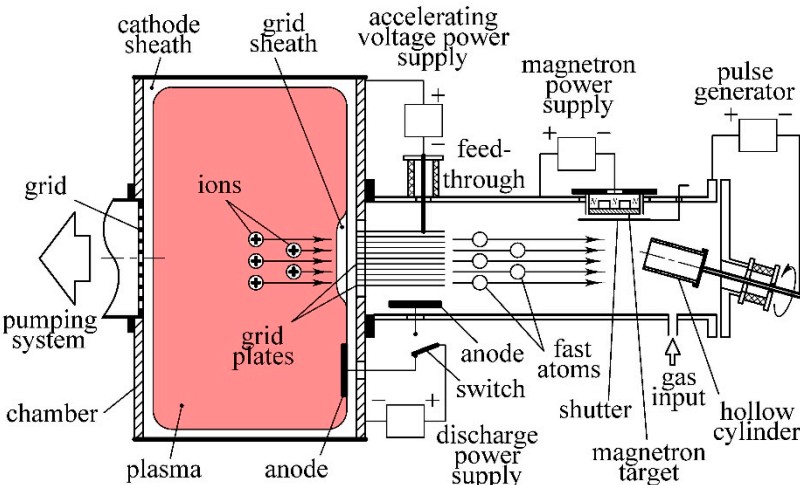

**Figure 1.** Schematic of experimental setup for the combined processing of complex-shaped parts.

The wall of the cylindrical chamber has a rectangular, 31-cm-wide, and 7-cm-high hole, covered by an accelerating grid composed of 11 plane-parallel 0.5-mm-thick, 300-mm-wide, and 150-mm-long titanium plates. Plates with 5.5-mm-thick inserts between them are fastened together with tie rods. The accelerating grid is attached to the chamber walls using four ceramic insulators protected from deposition of metal films by hollow cylindrical screens (Figure 2).

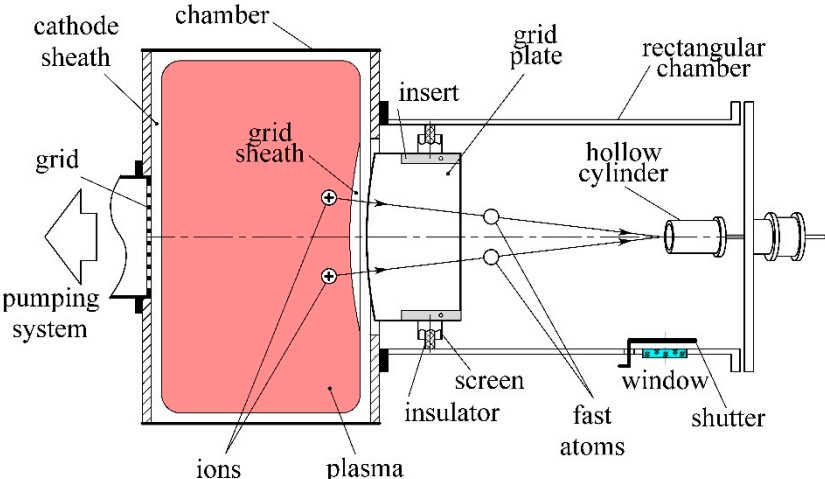

**Figure 2.** Schematic of experimental setup for the combined processing (top view).

At the top of the rectangular chamber, a magnetron [13,14] with 10-cm-diameter round target and a high-voltage feedthrough are mounted. The feedthrough allows to connect the grid to an accelerating voltage power supply and regulate the negative voltage from zero to 5 kV. The positive pole of the glow discharge power supply can be connected by means of a switch to either an anode inside a rectangular chamber or a second anode inside a cylindrical chamber.

The working gas (argon) is fed into a rectangular chamber and pumped out by rotary and turbomolecular pumps to a residual gas pressure of 0.001 Pa through a flat grounded grid on the left wall of the cylindrical chamber. The gas pressure in the chamber is measured by a Baratron gauge and adjusted from the control panel. The gas discharge power supply

and the accelerating voltage power supply are controlled from the same control panel. The grounded grid prevents plasma from entering the vacuum pumping system.

### 2.2. Filling the Chambers with Plasma

Switching on the power supply between the anode and the chamber at a gas pressure of 0.1 Pa leads to the ignition of a glow discharge with a voltage of $U$ ~400 V and the filling of the cylindrical chamber with a homogeneous plasma. At the indicated pressure, the mean free pass of argon atoms amounts to 7 cm [15] and the charge-exchange length of argon ions increases from 0.13 to 0.24 m as their energy grows from 0.5 to 6 keV [16,17]. Ionization of the gas in the chamber is carried out mainly by electrons emitted by the chamber. They are accelerated in the sheath near the chamber wall, fly through the plasma, and are reflected back into the plasma in the sheath near the opposite wall. These electrons can fly through the plasma hundred times, and the length of their path from the chamber wall to the anode $L = 4\, V/S_a$, where $V$ is the chamber volume and $S_a$ is the anode surface area [18,19], can exceed the chamber width by two orders of magnitude. At direct current, for example, $I = 2$ A, the discharge voltage $U$ does not depend on the gas pressure $p$ until it decreases to a critical value $p_o$. At the pressure $p_o$, the path $\Lambda$, during the passage of which the electrons emitted by the chamber spend all their energy $eU$ on gas ionization, increases to $L$. Each of them forms $N = eU/W$ ions, where $W$ is the gas ionization cost, for argon equal to $W = 26$ eV [20]. A comparable contribution to gas ionization is made by fast electrons formed in the sheath near the chamber wall.

When an accelerating voltage $U_a$ of up to 5 kV is applied to the grid, the width of the space charge sheath between the grid and the plasma increases [21]. Plasma ions accelerated in this sheath fly into the gaps between the grid plates. Due to the inhomogeneity of the electric field at the edges of the plates, they scatter at small angles. Therefore, the ions touch the side surfaces of the plates and, as a result, are converted into fast atoms of a beam with a 6-cm-high and 28-cm-wide cross-section injected into the rectangular chamber. The sections of the grid plates bordering the plasma emitter have the shape of a 60-cm-radius circle segment (Figure 2). Therefore, the trajectories of ions accelerated in the sheath between the plasma emitter and the grid, flying into the gaps between its plates, as well as the trajectories of fast atoms escaping from the grid after neutralization of the ion charges on the surfaces of the plates, are directed to the center of this circle.

The width of the convergent beam of accelerated particles, approximately equal to 28 cm near the plasma emitter in the cylindrical chamber, decreases by a factor of twenty at a distance of 3 cm from the center of the circle. As a result, the flux density of fast argon atoms in a beam bombarding a cylindrical sample increase by a factor of twenty compared to a beam without compression. Accordingly, the rate of the sample sputtering also increases by twenty times.

After connection of the glow-discharge power supply to the anode in the rectangular chamber and filling it with a homogeneous plasma, applying high-voltage pulses of negative polarity to a sample immersed in the plasma should lead to breakdowns between the plasma and protrusions on the rough surface of the sample. A voltage pulse generator was used with an amplitude of negative pulses adjustable from 3 to 30 kV and a pulse width adjustable from 5 to 50 μs, following one after another with a frequency adjustable from 1 to 50 Hz. The width of the pulse front is 5 μs. The maximum output power of the generator is 0.5 kW with a maximum power input of 1 kW.

To determine the amplitude of voltage pulses on a processed cylindrical sample, at which breakdowns occur between the plasma and protrusions on its surface, voltage oscillograms were recorded using a voltage divider and a GDS-72104 oscilloscope by GW Instek (Taiwan). For visual observation of breakdowns, there is a quartz window with a movable shutter on the side wall of the rectangular chamber that prevents from deposition of metal films.

A similar shutter protects the magnetron target at the top of the chamber, made of a heat-resistant CoCrMo alloy, from contamination. Preliminary tests showed that at an

argon pressure of 0.4 Pa and a stabilized current in the target circuit of 6 A, the rate of coating deposition on a flat substrate installed horizontally on the chamber axis amounts to 12 μm/h. To measure the thickness of the deposited coating, a mask in the form of a 0.5-mm-thick and 5-mm-wide titanium strip is preliminarily fixed on the substrate. After the coating is deposited, the mask is removed from the substrate and the thickness of the coating is determined as the step height on the profilogram of its surface. When dividing the thickness by the processing time, the deposition rate is obtained.

When the magnetron is turned on, the quartz window for visual observation is covered by a protective shutter, and after the magnetron is turned off, its target is also covered by another shutter. The quartz window allows to measure the sample temperature during its processing using an IMPAC IP 140 infrared pyrometer by LumaSense Technologies GmbH (Raunheim, Germany).

### 2.3. Instruments for Characterisation of the Samples

An industrial EOS M 400 additive manufacturing facility was used to produce the experimental samples. For selective laser melting technology, it is equipped with four lasers each of 04-kW power. Productivity amounts to 100 cm$^3$/h with 0.4 m × 0.4 m × 0.4 m dimensions of the chamber.

An optical 3D measuring system MicroCAD Premium plus manufactured by GFMesstechnik GmbH (Berlin, Germany) was used to measure the sample profiles. Its measuring volume amounts to 2.9 mm × 2.4 mm × 1.0 mm, the number of measured points 2452 × 2056 with measuring time of 7 s, Z resolution of 0.1 μm, and X—Y resolution of 1 μm.

A profilometer HOMMEL TESTER T8000 produced by Hommelwerke GmbH (Germany) was used to measure the surface roughness and coating thickness.

Measuring range/resolution:

±8 μm/1 nm; ±80 μm/10 nm; ±800 μm/100 nm; ±8000 μm/1000 nm.

Filter:

- cut-offs—0.025/0.08/0.25/0.8/2.5/8.0 (mm), selectable;
- tracing speed Vt—0.05/0.15/0.5 mm/s, or variable 0.01–2.0 mm/s in 0.01 increments;
- tracing length lt—0.48/1.5/4.8/15/48 mm or variable 0.1–200 mm;
- measuring length lm—0.40/1.25/4.0/12.5/40 mm or variable;
- cut-off λ (mm)—0.08/0.25/0.8/2.5/8.0;
- Gauss—digital filter (mm) cut-offs 0.025/0.08/0.25/0.8/2.5/8.0.
- Stylus tip radius 2 μm.
- Software TURBO ROUGHNESS, Hommel Map Expert 4.1.

An infrared pyrometer IMPAC IP 140 manufactured by LumaSense Technologies GmbH (Frankfurt am Main, Germany) was used to measure temperature of the samples under treatment:

| | |
|---|---|
| Temperature measurement range: | 200–1300 °C |
| Spectral range: | 2–2.8 μm |
| IR detector: | PbS |
| Radiation coefficient (ε): | 0.1 . . . 1.0 |
| Reproducibility: | 0.1% of the measured value °C + 1 °C |
| Dimensions (L × W × H): | 195 mm × 56 mm × 62.5 mm |

A Nanovea M1 Hardness and Scratch Tester produced by Nanovea Mechanical Testing (Irvine, CA, USA) was used to measure the coating microhardness and adhesion:

| | |
|---|---|
| Dimensions of the slide table, mm | 150 × 150 |
| Maximum Z-axis clearance, mm | 140 |
| Load range, N | 1–40 |
| Minimum noise level, min | 0.75 |
| Depth range, μm | 300 |
| Depth detection accuracy, nm | 10 |
| Scratching speed, mm/min | 0–240 |
| Lens magnification | 10×, 20×, 50× and 100× |

Sample Requirements:

Thickness                                      Exceeding the depth of indentation at least 10 times
Surface                                         plane-parallel
Roughness Ra not more than 0.32 μm.

A Calotest instrument manufactured by CSM Instruments (Peseux, Switzerland) was used to evaluate the abrasive wear resistance of the samples. The dimensions of its work table are 80 mm × 80 mm (fixing samples by means of 2 clips). The shaft rotation speed is 10–1000 rpm (with the possibility of continuous adjustment by a potentiometer), and the standard diameter of the ball amounts to 20, 25.4, or 30 mm. The set time for wear ranges from 1 to 999 s.

## 3. Results

### 3.1. Explosive Ablation of Superficial Protrusion

After installation of a hollow cylindrical sample with a diameter of 6 cm and a length of 8 cm on a holder in a rectangular vacuum chamber of the setup (Figure 1), it was evacuated to a residual gas pressure of 0.001 Pa, and a pressure of 0.5 Pa was set by supplying argon. The discharge power source was connected to the anode of the rectangular chamber and a voltage of ~500 V was applied to it. As a result, the chamber was filled with a uniform glow discharge plasma. However, when high-voltage pulses were applied to the sample immersed in the plasma, no breakdowns between it and the plasma were observed at any parameters of the pulses.

Studies have shown that explosive processes on the surface of the sample weakly depend on the discharge current and gas pressure in the chamber. The main factor affecting them is the width of the leading edge of high-voltage pulses. It was 5 μs for the generator used, and in order to significantly reduce the width of the pulse front, it had to be abandoned.

Further, pulses on the sample were formed during the breakdowns of a spark gap between the ground and the positive plate of a capacitor with a capacity of 0.02 μF (Figure 3). The negative plate of the capacitor is connected through a low-inductance resistor to the ground, and through a high-voltage diode $D_2$, to a hollow cylindrical sample immersed in the plasma. For a time of ~0.02 s, the capacitor is charged from a high-voltage transformer through a resistor $R_1$ and a high-voltage diode $D_1$ to a spark gap actuation voltage of 10 kV, the capacitor discharge current pulse through the low-inductance resistor $R_2$ forms a high-voltage pulse between the sample and the plasma. In the event of a breakdown between them, the capacitor discharge current switches to the sample circuit, and a high-voltage diode in this circuit prevents the current from changing direction in the second half-cycle of oscillations. Therefore, almost all of the energy stored in the capacitor is spent on the destruction of microscopic protrusions on the surface of the sample.

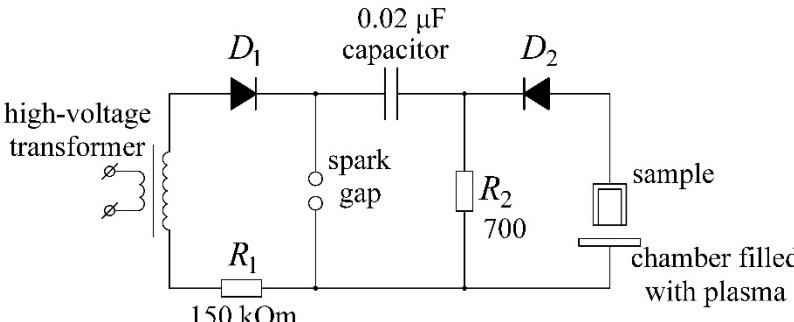

**Figure 3.** Scheme for generation of high-voltage pulses between the sample and plasma.

The oscillograms of voltage pulses on the sample in the absence of plasma in the chamber at a pressure of 0.002 Pa showed that, with the indicated circuit parameters, the width of the leading edge of the pulse is 0.5 μs, which is 10 times less than when using a high-voltage pulse generator. After filling the chamber with argon to a pressure of 0.5 Pa and applying a voltage of ~500 V to the anode, the chamber was filled with a uniform glow

discharge plasma, and bright sparks appeared on the surface of the sample immersed in the plasma, similar to the cathode spots of a vacuum arc [22,23]. A minute later, the window on the chamber wall darkened and had to be covered with a shutter.

After treatment of an 8-cm-long hollow cylindrical sample made of Co-Cr-Mo for 30 min with pulses at a repetition rate of ~40 Hz, its surface became smooth and matte. There were no large protrusions on it, and the roughness of both the outer and inner surfaces measured using a HOMMEL TESTER T8000 profilometer by Hommelwerke GmbH (Germany) ranged from Ra = 0.9 μm to Ra = 1.2 μm. The porosity was almost completely eliminated. Hence, the microexplosions not only destroy protrusions on the sample surface, but also, as a result of surface melting, eliminate micropores on it. According to the experimental results, the energy stored in a 0.02 μF capacitor charged to 10 kV, approximately equal to 2.5 J, can be considered the optimal value of the pulse energy.

Explosive ablation of protrusions on the surface of a sample immersed in plasma is an analog of electroerosive grinding, which uses electroerosive metal destruction. It essentially differs in that there is no dielectric medium or any electrode tool, the approach of which to the sample surface should lead to a breakdown between them with the formation of a dense plasma channel and local destruction of the sample surface. In our case, breakdowns between the plasma and the sample immersed therein are similar to the formation of plasma torches on the cathode surface of electron guns with explosive electron emission and are determined by the size and location of microscopic protrusions on the sample surface.

### 3.2. Polishing with a Beam of Fast Argon Atoms

Beam polishing is effective only at large angles of incidence of fast atoms on the treated surface $\alpha$ = 75–85°. In the case of flat substrates, it is quite simple to create the necessary processing conditions. It is only necessary to install the substrate on a holder rotating in the vacuum chamber in such a way that the angle of incidence of fast atoms on its surface is equal to the specified value.

For products of complex shape with cavities, polishing at large angles of incidence of fast atoms on the surface to be treated $\alpha$ = 75–85° requires new technical solutions. For example, an 8-cm-long hollow cylindrical sample, with an outer diameter of 6 cm and an inner diameter of 5 cm, after explosive ablation of protrusions on its surface for 30 min, was treated with a ribbon beam of fast argon atoms. For this, the positive pole of the discharge power source was disconnected from the anode of the rectangular chamber using a switch and connected to the anode of the cylindrical chamber. When a voltage $U_d$ = 500 V is applied to it, the chamber is filled with a homogeneous glow discharge plasma at an argon pressure of 0.1 Pa.

After a voltage $U$ = 5 kV is applied to the plates of the accelerating grid, ions from the plasma are accelerated by a voltage $U + U_d$ = 5.5 kV in the ion space charge sheath between the plasma and the grid plates and fly into the gaps between the plates. Since the sections of the plates bordering the plasma emitter have the shape of a segment of a circle with a radius of 60 cm (Figure 2), the trajectories of ions accelerated in the sheath between the plasma emitter and the grid and flying into the gaps between the plates, as well as the trajectories of fast atoms flying out of the grid after neutralization of the ion charges on the surfaces of the plates, are directed towards the center of this circle, located near the inlet of the cylindrical sample.

Compression of the beam over the width of its cross-section from 26 cm to 1 cm (Figure 2) at a constant cross-section height of 6 cm (Figure 1) allows simultaneous polishing of the inner and outer surfaces of the hollow cylindrical sample with a ribbon beam. At an angle of 15° between the axis of the sample and the axis of the chamber, the upper half of the beam cross-section sputters at an angle of incidence of 75° the upper part of the inner surface of the rotating cylinder, and the lower half of the beam cross-section sputters at an angle of incidence of 75° the lower part of its outer surface (Figure 1).

The surface roughness of a hollow cylindrical sample was measured using a HOMMEL TESTER T8000 profilometer from Hommelwerke GmbH (Germany). To determine the

coating thickness on the sample, masks were applied to its outer and inner surfaces before coating deposition. After the coating deposition and mask removal, the coating thickness was determined by the step height on the surface profilogram.

Polishing a sample with a beam consists in removing protrusions from its surface as a result of their sputtering by fast atoms. When a rotating sample is irradiated by a beam of fast atoms with a large angle of incidence on its surface (about 75°), fast atoms first of all sputter the tops of protrusions on the surface of the sample, while the recesses between them remain in shadows. To ensure a low surface roughness, it is necessary to lower the level of the protrusions to the level of the recesses. To do this, you need to remove a large amount of material from the surface and spend a lot of time. With an increase in the initial roughness, the polishing time of the sample increases noticeably.

After measuring, using a HOMMEL TESTER T8000 profilometer, the initial surface roughness Ra = 1.2 μm of one of the samples subjected to explosive ablation of surface protrusions, it was mounted back on the holder (Figure 1) and the surface of the rotating at a speed of 60 rpm sample was sputtered for an hour at an angle of incidence $\alpha = 75°$ by 5.5-keV argon atoms with a beam current of 1 A and a gas pressure $p = 0.2$ Pa. The sample cooled in vacuum was removed from the chamber and the roughness of its inner surface Ra = 0.6 μm and outer surface Ra = 0.4 μm was measured. The sample was then placed back on the holder and sputtered for another two hours. As a result, the roughness of the inner surface of the sample decreased to Ra = 0.08 μm, and the outer surface to Ra = 0.06 μm (Figure 4).

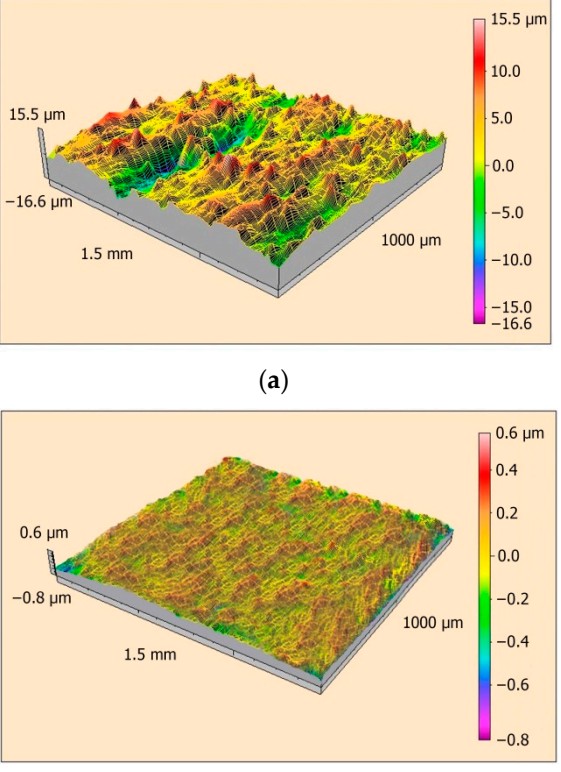

(**a**)

(**b**)

**Figure 4.** 3D profilograms of the outer surface sample before processing Ra = 5 μm (**a**) and after explosive ablation and polishing with a beam of fast atoms Ra = 0.06 μm (**b**).

### 3.3. Coating Deposition

The possibility of reducing the surface roughness of a sample by depositing a coating of the sample material in the recesses between its protrusions has been discussed in the literature. To carry out an experimental verification of this assumption, a sample subjected to explosive ablation of surface protrusions with a surface roughness Ra = 1.1 μm and

masks on its outer and inner surfaces was mounted on a rotating holder. At an argon pressure of 0.2 Pa, a magnetron discharge was switched on with a stabilized current 4 A of ions sputtering a magnetron target made of CoCrMo alloy [24,25], and a coating of the sample material was deposited for 3 h.

After removing the coated sample from the chamber and removing the masks from its surface, the average coating thickness was determined from the height of the steps of the profilograms, which amounted to 15 μm. Despite the fact that it was an order of magnitude higher than the height of microscopic protrusions on the original surface of the sample, the surface roughness of the coating amounted to Ra = 0.9 μm, i.e., it practically did not change.

This means that the deposition of a coating with a thickness noticeably exceeding the height of the surface protrusions cannot give a polishing effect. This is because the rate of the coating deposition on the protrusions of the surface exceeds the rate of deposition in the recesses between them. Therefore, the surface topography after coating deposition practically does not change.

To reduce the polishing time for samples made by the additive method, it is possible to deposit a coating of the same material on the surface of the sample when sputtering the tops of protrusions (Figure 5). In this case, fast atoms sputter only the tops of protrusions on the surface, while recesses remain in the shadows and the coating is freely deposited in the recesses.

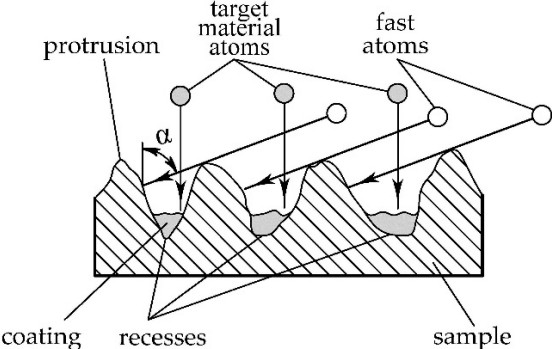

**Figure 5.** Scheme of sputtering by fast argon atoms of protrusions on a rough surface with simultaneous coating deposition. α—the angle of fast atoms incidence to the surface.

One of the cylindrical samples subjected to explosive ablation of protrusions with a surface roughness of Ra = 1.0 μm was mounted on a holder (Figure 1) and the surface of the sample rotating at a speed of 60 rpm was sputtered for an hour at an angle of incidence α = 75° with 5.5-keV argon atoms at beam current $I$ = 1 A and gas pressure $p$ = 0.2 Pa. Simultaneously, a coating was deposited on the sample at a stabilized current of 4 A in the circuit of the magnetron target made of CoCrMo alloy.

The sample cooled in vacuum was removed from the chamber and the roughness of its inner surface Ra = 0.3 μm and outer surface Ra = 0.2 μm was measured. Then the sample was again installed on the holder and sputtered for another two hours with simultaneous deposition of the coating. As a result, the roughness of the inner surface of the sample decreased to Ra = 0.02 μm, and that of the outer surface to Ra = 0.01 μm.

To demonstrate the possibility of synthesizing wear-resistant coatings on parts fabricated by the additive method, the CoCrMo magnetron target was replaced with a titanium target. A cylindrical sample with a minimum roughness of the inner surface Ra = 0.02 μm and the outer surface Ra = 0.01 μm was placed on the holder. Masks were preliminarily applied to both surfaces in order to further measure the thickness of the synthesized coating.

After the chamber was evacuated, in order to clean and activate the sample surface, it was sputtered for 5 min with 5.5-keV argon atoms at an equivalent beam current $I$ = 1 A and an incidence angle of 75°. Then, a mixture of argon with 20% nitrogen was fed into the chamber, and at a gas pressure of 0.2 Pa and a current of 4 A in the titanium target circuit, a coating was deposited for an hour on the rotating sample. During the deposition process,

the coating was bombarded at an angle of incidence of 75° by fast atoms with an energy of 2.5 keV and an equivalent current $I = 0.5$ A.

According to the mutual arrangement of the magnetron target and the hollow cylindrical sample indicated in Figure 1, less than half of the sputtered target atoms were deposited on its surface. However, sputtered atoms arrived at the outer and inner surfaces in approximately equal amounts. This is evidenced by the results of measuring the thickness of the coatings according to the profilograms of the steps on the surface formed after the removal of the masks (Figure 6).

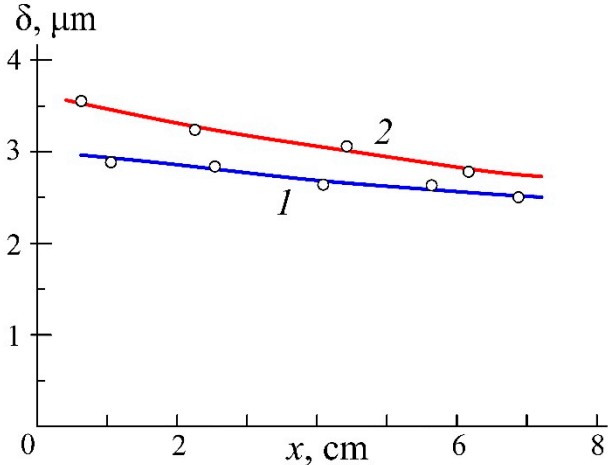

**Figure 6.** Dependence of the coating thickness on the inner (1) and outer (2) surfaces of the hollow cylinder on the distance from its end.

On the inner surface, the coating thickness near the end of the cylinder closest to the magnetron target (see Figure 1) is equal to 2.9 μm and decreases monotonically to 2.5 μm near its opposite end (Figure 6). On the outer surface, the coating thickness near the end of the cylinder is equal to 3.6 μm and decreases monotonically to 2.8 μm at the opposite end of the sample.

Bombardment of a growing coating by energetic ions can drastically improve the coating properties [26–28]. For convenience in measuring the microhardness of the synthesized coating and evaluating its adhesion, 3-cm-long and 6-mm-wide fragments were cut from an 8-cm-long and 6-cm-diameter sample along the cylinder generatrix.

To evaluate the abrasive wear resistance of a cylindrical CoCrMo sample and its dependence on the treatment with fast argon atoms and deposition of a wear-resistant coating, a Calotest instrument manufactured by CSM Instruments (Peseux, Switzerland) was used. A rotating ball was placed on a fragment cut from the sample with a load of 0.2 N, and an abrasive suspension was fed into the contact zone. Abrasive particles in the contact zone and the applied external force led to local abrasion of the sample surface. When the ball rotates, a spherical wear-resistant recess is formed on the surface of the sample. As the ball rotates, it produces a spherical wear notch on the sample surface. The diameter d of the notch is determined using an optical microscope. The ball radius R = 12.5 mm appreciably exceeds d and the volume of worn material is equal to V = π·d4/64R. Figure 7 presents dependence of the abrasion volume on the test time for a not-processed sample (I) and sample coated with TiN, after the combined treatment (II). This shows that, due to the treatment of the sample with explosive ablation of surface protrusions, polishing with a beam of fast neutral atoms, and deposition of wear-resistant coating, the abrasive wear became three times smaller.

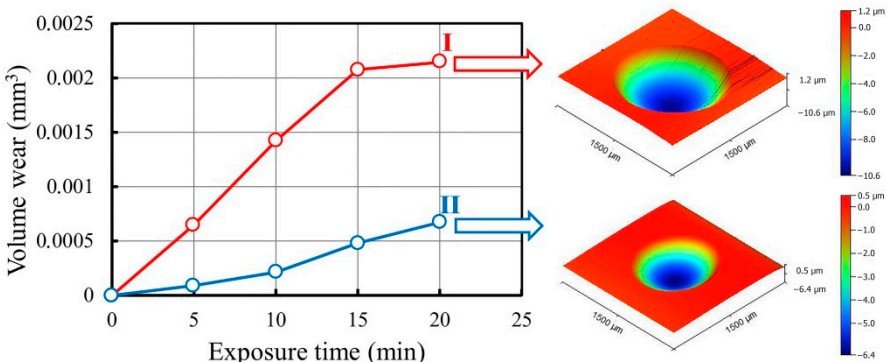

**Figure 7.** Dependencies of the abrasion volume on the test time for a not-processed sample (I) and a sample after combined treatment (II).

X-ray diffraction phase analysis was used to determine the phase composition of the coating. X-ray diffraction patterns were taken using an Empyrean Series 2 X-ray diffractometer with monochromatized CuKα radiation produced by PANALYTICAL (Netherlands) (Figure 8). The method of a grazing incident X-ray beam made it possible to establish the presence of a single TiN phase in the coating. This is in good agreement with the gold color of the coating. Moreover, it was found that the coating has a strong texture.

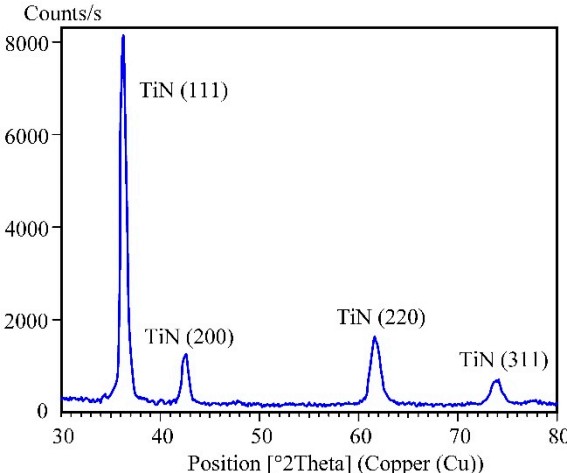

**Figure 8.** The X-ray diffraction pattern of the synthesized coating taken using the method of a grazing incident X-ray beam.

A Hardness and Scratch Tester produced by Nanovea Mechanical Testing (Irvine, CA, USA) was used to measure characteristics of the coating synthesized on the sample. The coating microhardness amounted to 2150 HV40. The first critical load to result in the appearance of an acoustic emission and first cracks on the coating amounted to 18 N. Thus, quite good adhesion was achieved due to the surface activation by the fast argon atoms before the coating synthesis and the coating bombardment by the fast atoms during its deposition.

## 4. Discussion

To improve the surface quality of parts produced using the additive manufacturing technology, at least three disadvantages must be eliminated: the surface porosity, its roughness, and inconsistency of the surface layer material with the conditions of the part application. Processing the surface of a part obtained by additive manufacturing with high-current pulsed electron beams makes it possible to almost completely get rid of porosity and reduce the surface roughness Ra from tens to units of micrometers [4]. The main factors are melting the surface protrusions and a rapid subsequent recrystallization. In any case,

the surface processed with a broad electron beam must be further polished to provide a higher surface finish class.

In our case, melting of the surface protrusions takes place during their explosive ablation and covers numerous pores on the sample surface. After treatment of an 8-cm-long hollow cylindrical sample for 30 min with pulses at a repetition rate of ~40 Hz, its surface became smooth and matte. There were no large protrusions on it, and the roughness of both the outer and inner surfaces ranged from Ra = 0.9 μm to Ra = 1.2 μm. The porosity was almost completely eliminated. This means that microexplosions not only destroy protrusions on the sample surface, but also eliminate micropores as a result of surface melting. According to the experimental results, the energy stored in a 0.05 μF capacitor charged to 10 kV, approximately equal to 2.5 J, can be considered as the optimal value of the pulse energy.

As polishing with a beam is effective only at large angles of incidence of fast atoms on the treated surface $\alpha = 75$–$85°$, polishing of complex-shape parts with cavities requires new technical solutions. For example, an 8-cm-long hollow cylindrical sample with an outer diameter of 6 cm and an inner diameter of 5 cm after explosive ablation of protrusions on its surface was treated with a ribbon beam of fast argon atoms allowing simultaneous polishing of the inner and outer surfaces of the sample. At an angle of 15° between the axis of the rotating sample and the axis of the chamber, the upper half of the beam cross-section sputters at an angle of incidence of 75° in relation to the upper part of the inner surface of the rotating sample, and the lower half of the beam cross-section sputters at an angle of incidence of 75° in relation to the lower part of its outer surface (Figure 1).

The investigations showed that polishing the inner and outer surfaces of a complex-shaped part (cylinder) with fast argon atoms allows for the reduction of the surface roughness from Ra ~ 1.2 μm to Ra ~ 0.05 μm within 3 h.

The polishing time appreciably diminishes when sputtering of the protrusion tops by fast argon atoms under a large angle of incidence to the sample surface is accompanied with deposition on the surface of coating made of the sample material. When a hollow cylindrical sample was subjected to simultaneous coating deposition and sputtering by fast argon atoms under large angle of incidence to the sample surface, its roughness diminished within 3 h from Ra ~ 1.0 μm to Ra ~ 0.01 μm.

Due to polishing, the sample surface wear-resistant titanium nitride coating is characterized with good adhesion at the thickness of 3.5 μm and microhardness of 2150 HV40. Due to the combined treatment of the sample with explosive ablation of surface protrusions, polishing with a beam of fast neutral atoms, and deposition of wear-resistant coating, the abrasive wear became three times smaller.

It can be concluded that all scientific tasks noticed in Section 1 have been fulfilled. The outlook for further research should be the implementation of the obtained results for the production of some parts in industry.

## 5. Conclusions

1. The study of polishing a part with a beam of fast argon atoms with a large angle of incidence on the part surface showed that, with the help of the beam, it is possible to significantly increase the surface finish class.
2. Deposition on a part of a coating with a thickness significantly exceeding the height of the protrusions on its surface does not reduce its roughness.
3. A simultaneous deposition of coating on a part surface being sputtered by a beam of fast argon atoms with a large angle of incidence to the surface noticeably increases the polishing speed.
4. Due to combined treatment of the part with explosive ablation of surface protrusions, polishing with a beam of fast neutral atoms, and deposition of wear-resistant coating, the coating adhesion was substantially improved and the abrasive wear became three times smaller.

**Author Contributions:** Conceptualization, A.M., M.V. and S.G.; methodology, A.M. and M.V.; software, E.M.; validation, A.M., M.V. and Y.M.; formal analysis, Y.M.; investigation, E.M. and Y.M.; resources, E.M. and Y.M.; data curation, M.V. and Y.M.; writing—original draft preparation, A.M. and M.V.; writing—review and editing, A.M. and S.G.; visualization, E.M.; supervision, A.M. and S.G.; project administration, M.V.; funding acquisition, A.M. All authors have read and agreed to the published version of the manuscript.

**Funding:** This work was supported financially by the Ministry of Science and Higher Education of the Russian Federation (project No FSFS-2021-0006).

**Institutional Review Board Statement:** Not applicable.

**Informed Consent Statement:** Not applicable.

**Data Availability Statement:** Data sharing is not applicable to this article.

**Acknowledgments:** The study was carried out on the equipment of the Center of collective use of MSUT "STANKIN" supported by the Ministry of Higher Education of the Russian Federation (project No. 075-15-2021-695 from 26 July 2021, unique identifier RF 2296.61321X0013).

**Conflicts of Interest:** The authors declare no conflict of interest.

## Abbreviations

| | |
|---|---|
| d | Diameter of the notch, mm |
| $I$ | Discharge current, A |
| $L$ | Length of electrons path from the chamber wall to the anode, m |
| $l$ | Evaluation width, mm |
| $lt$ | Tracing length, mm |
| $ln$ | Evaluation length, mm |
| $lr$ | Sampling length, mm |
| $p$ | Gas pressure in the chamber, Pa |
| $p_o$ | Critical pressure, Pa |
| $Ra$ | Arithmetical mean deviation of the roughness profile, µm |
| $r_{tip}$ | Stylus tip radius, µm |
| $S_a$ | Anode surface area, $m^2$ |
| $U$ | Grid plates voltage, V |
| $U_a$ | Accelerating voltage, V |
| $U_d$ | Discharge voltage, V |
| $V$ | Chamber volume, $m^3$ |
| $v_t$ | Tracing speed, mm/s |
| $W$ | Gas ionization cost, eV |
| $\alpha$ | Angle of incidence of fast atoms on the treated surface, ° |
| $\Lambda$ | Path, during the passage of which the electrons emitted by the chamber spend all their energy, m |
| $\lambda c$ | Long-wave profile filter, mm |

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
