# Peer review of "Combined Treatment of Parts Produced by Additive Manufacturing Methods for Improving the Surface Quality"

_technologies, doi:10.3390/technologies10060130_

Round 1

Reviewer 1 Report

Overall, the work is solid. I also appreciate the detailed documentation of experimental conditions. A few comments are as follows:

1.      Need a table to summarize the parameters for all the experiments discussed in the paper. It is inconvenient to search around the paper for numbers, such as angles, voltage, amps, and times, when making comparisons.

2.      Line 348-349: “On the inner surface, the coating thickness near the end of the cylinder is equal to 2.9 μm and decreases monotonically to 2.5 μm at its other end”. I am confused about “the end” and “other end”. I cannot tell which ends the authors are referring to. Please be more specific. It may be helpful to point them out in the figure.

3.      Line 373 and 436: The expression “decreased three times” is meaningless. How much is a one-time decrease? The authors could use expressions such as “decreased by 70%” or “decreased by 2/3” instead. 

Author Response

First of all, we would like to thank you for your encouraging assessment of our research: “Overall, the work is solid. I also appreciate the detailed documentation of experimental conditions”.

You are perfectly right, it was very difficult to understand the meaning of the sentence: “On the inner surface, the coating thickness near the end of the cylinder is equal to 2.9 μm and decreases monotonically to 2.5 μm at its other end”. In the revised manuscript we changed it as follows: “On the inner surface, the coating thickness near the end of the cylinder closest to the magnetron target (see Figure 1) is equal to 2.9 μm and decreases monotonically to 2.5 μm near its opposite end”. On your advice we changed Figure 1 and indicated the hollow cylinder there.

On your advice we replaced the meaningless expression “decreased three times” with simpler and more understandable “became three times smaller”.

Thank you very much for cooperation.

Reviewer 2 Report

The treatment process of the sample surface by the author is similar to the process of removing surface impurities in the preparation stage of the high-energy plasma immersion ion implantation process

The surface roughness of the sample is reduced by using glow discharge method to remove the bulges on the surface of the sample, and the all-orientational polishing is realized by using a similar plasma discharge device. Generally speaking, it is a good process and worth using

The only question is: how to improve the processing efficiency?

Author Response

We agree with you that our treatment process of the sample surface is similar to the process of removing surface impurities in the preparation stage, no matter, of the coating deposition, or plasma immersion ion implantation, or some other processes.

The previously used glow discharge method to remove the bulges on the surface of the sample was not effective and you could hardly have a polishing effect. To answer your question “how to improve the processing efficiency?”, we proposed a new approach: first, polishing with fast argon atoms at a large angle of incidence on the sample surface and, secondly, simultaneous deposition of coating on the sample surface being polished. This processing made it possible to get rid of porosity and reduce the surface roughness from Ra ~ 5 µm to Ra ~ 0.05 µm.

You concluded “it is a good process and worth using”. Thank you very much for your encouraging assessment of our research.

Reviewer 3 Report

Dear author(s), please find some comments on the manuscript ‘Combined Treatment of Parts Produced by Additive Manufacturing Methods for Improving the Surface Quality’, Manuscript ID: technologies-2066823:

1.      Considering the “Introduction’ section is well-written and interesting for a regular reader. However, some issues must be raised that there is a lack of critical review of the current state of knowledge around the presented topic.

2.      The motivation of the work is found, nevertheless, it does not respond to the critical review is lost. Please try to emphasize the novelty by presenting critical issues.

3.      Some values of the experiment coefficients were not justified in section 2.1. From that matter, it looks like selected arbitrarily.

4.      As a supplement to the previous comment, any information on the roughness measurement was not provided. Lines, 66-68 and 167-168 were only the equipment introduced. Any details? Any measuring parameters?

5.      Considering the measurement of surface roughness, further, there is no word again the validity of the technique (device) applied. Especially, there is no introduction to both measurement uncertainty and noise. Please look for some examples:

(1)   https://doi.org/10.1088/2051-672X/3/3/035004

(2)   https://doi.org/10.3390/coatings12060726

(3)   https://doi.org/10.3390/app7010054

6.      Concluding with the previous suggestion, each of the six instruments (presented in section 2.3.) was not introduced appropriately. More details are urgently required.

7.      As in comment no.3, the angle of 75° (line 271) was not justified. Why not 65° or 85°?

8.      In Figure 4 the 3D (areal) surface roughness was presented so, respectively, it was not introduced, why a Ra (profile) roughness was considered? It is often proposed, especially in engine (cylindrical) surfaces, to analyse the profile (Ra) parameters instead of an areal (Sa). However, it must be appropriately presented.

9.      In section no.4, Discussion, there is missing a critical review of the result presented. It looks more like a pre-conclusion section. Please try to emphasize what was not received against the results expected.

10.  Reflecting on the previous issue, ‘The outlook’ would be required. The author (s) should present some further ideas or take some of its advantages in the section ‘Discussion’.

11.  The ‘Conclusion’ section must be improved. In fact, the most encouraging and, respectively, important conclusions were presented. Nevertheless, some elaboration on the main issues must be provided.

12.  There are many shortcuts and parameters that, in some cases, the reader feels confused about. An additional section, e.g. Abbreviation, should be included.

From the manuscript, it looks really interesting but the reader seems to be lost in some cases, unfortunately. So, respectively, the suggested aspects must be significantly improved in the further manuscript consideration to be published in the quality journal as the Technologies is.

Author Response

  1. Considering the “Introduction” section is well-written and interesting for a regular reader. However, some issues must be raised that there is a lack of critical review of the current state of knowledge around the presented topic.

As a matter of fact, on the base of Refs [1-10] we presented in Lines 29-55 an adequate review of the methods for the surface polishing: mechanical polishing, electrochemical etching, laser-plasma polishing, polishing with a high-current pulsed electron beam, the ion beam polishing and polishing with a beam of fast neutral atoms.

  1. The motivation of the work is found, nevertheless, it does not respond to the critical review is lost. Please try to emphasize the novelty by presenting critical issues.

Presenting the polishing methods in Lines 29-55 we noticed their disadvantages thus emphasizing the novelty of our new approach.

  1. Some values of the experiment coefficients were not justified in section 2.1. From that matter, it looks like selected arbitrarily.

Regrettably, we could not find any “experiment coefficients” in section 2.1?

  1. As a supplement to the previous comment, any information on the roughness measurement was not provided. Lines, 66-68 and 167-168 were only the equipment introduced. Any details? Any measuring parameters?

On your advice, we presented in Lines 71-73 and Table 1 of the revised manuscript all details and measuring parameters for profilometer HOMMEL TESTER T8000 produced by Hommelwerke GmbH (Germany)

  1. Considering the measurement of surface roughness, further, there is no word again the validity of the technique (device) applied. Especially, there is no introduction to both measurement uncertainty and noise.

Since the measured surface area has rather large dimensions of 1.5 mm × 1 mm, so that the measurement time would have a reasonable value of about 4 hours, the following filters were selected:

- cut-off 0.25 mm;

- Gaussian filter 0.08 mm.

  1. Concluding with the previous suggestion, each of the six instruments (presented in section 2.3.) was not introduced appropriately. More details are urgently required.

On your advice we substantially enlarged information on the six instruments in section 2.3.

  1. As in comment no.3, the angle of 75° (line 271) was not justified. Why not 65° or 85°?

It is well-known that sputtering yield of argon ions with energies exceeding 1keV grows with increasing the angle of incidence on the sample surface starting from about 60°, reaches a maximum at 75°-80° and then abruptly diminishes (see, for instance, H. Gnaser. Energy and Angular Distributions of Sputtered Species. Topics in Applied Physics 2007, 110, 231-328, https://doi.org/10.1007/978-3-540-44502-9, Fig. 67 on page 105 for Al, Fig. 68 on page 106 for Ti and Fig. 70 on page 108 for Ni). The available data justify the choice of 75° for a maximum spattering rate.

  1. In Figure 4 the 3D (areal) surface roughness was presented so, respectively, it was not introduced, why a Ra (profile) roughness was considered? It is often proposed, especially in engine (cylindrical) surfaces, to analyse the profile (Ra) parameters instead of an areal (Sa). However, it must be appropriately presented.

In the ideal case (when each profilogram starts with the exact value of the height of the measured surface area), the parameter Sa (arithmetic mean of the absolute distance of surface points from the median plane) should be used. But, as a rule, for long lengths of profilograms (in our case, 1.5 mm), the height of the starting point of the next profilogram has a random spread relative to the actual value. Thus, roughness measurements along a line perpendicular to the stylus movement are too coarse compared to measurements along the stylus movement. The software of our HOMMEL TESTER T8000 profilometer allows to make no more than 201 profilograms to build a 3D surface and makes it possible to find the average value of Ra over all profilograms of the measured surface area of 1.5 mm × 1 mm. In our measurements, the resolution between points along the movement of the stylus is 1 µm, and in the transverse direction, 5 µm. In our case, the Ra parameter is very accurately determined as the average value of 201 profilograms, the resolution of each of them being one point per micron.

  1. In section no.4, Discussion, there is missing a critical review of the result presented. It looks more like a pre-conclusion section. Please try to emphasize what was not received against the results expected.

We indicated in “Discussion” that all scientific tasks noticed in Introduction have been fulfilled.

  1. Reflecting on the previous issue, ‘The outlook’ would be required. The author (s) should present some further ideas or take some of its advantages in the section ‘Discussion’.

In the section "Discussion", we noted that the perspective of our study should be the implementation of its results for the production of some parts in industry.

  1. The ‘Conclusion’ section must be improved. In fact, the most encouraging and, respectively, important conclusions were presented. Nevertheless, some elaboration on the main issues must be provided.

On your advice we revised the section “Conclusions” and enlarged the number of the main points.

  1. There are many shortcuts and parameters that, in some cases, the reader feels confused about. An additional section, e.g. Abbreviation, should be included.

On your advice we prepared and included the section “Abbreviation”.

We appreciate your work and time spent on reviewing our manuscript and hope that all corrections we made on your advice appreciably improved its quality.

Thank you very much.

Round 2

Reviewer 3 Report

Dear author(s), the manuscript titled ‘Combined Treatment of Parts Produced by Additive Manufacturing Methods for Improving the Surface Quality’, Manuscript ID: technologies-2066823, has been improved in a required manner so, respectively, can be further processed by the Materials journal.

Thank you for your full responses that, in their current form, were addressed properly and make the manuscript suitable for publication in a quality journal as the Technologies is.